# Real-World Experience of Olaparib Maintenance in High-Grade Serous Recurrent Ovarian Cancer Patients with *BRCA1/2* Mutation: A Korean Multicenter Study

**DOI:** 10.3390/jcm8111920

**Published:** 2019-11-08

**Authors:** E Sun Paik, Yong Jae Lee, Jung-Yun Lee, Wonkyo Shin, Sang-Yoon Park, Se Ik Kim, Jae-Weon Kim, Chel Hun Choi, Byoung-Gie Kim

**Affiliations:** 1Department of Obstetrics and Gynecology, Samsung Medical Center, Sungkyunkwan University School of Medicine, Seoul 06351, Korea; espaik@naver.com (E.S.P.); chelhun.choi@samsung.com (C.H.C.); 2Department of Obstetrics and Gynecology, Institute of Women’s Life Medical Science, Yonsei University College of Medicine, Seoul 03722, Korea; SVASS@yuhs.ac (Y.J.L.); yodrum682@gmail.com (J.-Y.L.); 3Center for Uterine Cancer, Research Institute and Hospital, National Cancer Center, Goyang 10408, Korea; 12958@ncc.re.kr (W.S.); parksang@ncc.re.kr (S.-Y.P.); 4Department of Obstetrics and Gynecology, Seoul National University College of Medicine, Seoul 03080, Korea; seikky@naver.com (S.I.K.); kjwksh@gmail.com (J.-W.K.)

**Keywords:** olaparib, recurrent ovarian epithelial carcinoma, maintenance therapy, treatment efficacy, adverse effects

## Abstract

Background: Olaparib maintenance therapy has shown efficacy and tolerability in patients with platinum-sensitive, high-grade serous recurrent ovarian cancer (HSROC) with *BRCA1/2* mutation (*BRCA*m). Our aim was to present real-world experience with olaparib in Korea. Method: We included HSROC patients with *BRCA*m treated with olaparib maintenance at four institutions in Korea between 2016 and 2018. Medical records were reviewed for clinico-pathologic characteristics, objective response, survival outcomes, and safety. Results: One hundred HSROC patients with *BRCA*m were included. *BRCA1* mutation was present in 71 patients (71.0%), and *BRCA2* mutation was present in 23 patients (23.0%). In terms of the best objective response with olaparib maintenance in 53 patients with partial remission from most recent chemotherapy, complete remission occurred in 12 (22.6%) and partial remission in four (7.5%), while 33 patients (62.3%) had stable disease. The 24 month progression-free survival was 42.4%, and 24 month overall survival was 82.1%. Grade 3 or more adverse events were as follows: anemia in 14 patients (14.0%), neutropenia in seven patients (7.0%), thrombocytopenia in two patients (2.0%), oral mucositis in one patient (1.0%), and soft tissue infection in one patient (1.0%). Conclusions: The safety and effectiveness of olaparib maintenance treatment in a real-world study were consistent with those reported in previous clinical trials.

## 1. Introduction

Ovarian cancer is the second most common cause of death related to gynecologic malignancy and the eighth leading cause of death from cancer in women worldwide [1]. The progression-free survival (PFS) of patients with advanced ovarian cancer after first-line chemotherapy is 4–12 months [2]. For ovarian cancer patients who respond to platinum-based chemotherapy at recurrence, a median PFS of 5–6 months from end of treatment has been reported [3,4]. Next-line chemotherapy can be offered to patients, but the treatment-free interval usually shortens each time recurrence is treated. Toxicity due to chemotherapy and acquired drug resistance are barriers to further treatment in patients with recurrent ovarian cancer, and tolerable treatment options for long-term disease control are needed. 

Olaparib (Lynparza™) is a poly (adenosine diphosphate–ribose) polymerase (PARP) inhibitor that induces synthetic lethality in *BRCA1/2*-deficient tumor cells [5,6]. Previous studies have shown the effectiveness of olaparib in platinum-sensitive high-grade serous recurrent ovarian cancer (HSROC). Data from a Phase II trial to assess the efficacy and safety of olaparib maintenance monotherapy in platinum-sensitive HSROC patients, Study 19 (NCT00753545), showed a significant improvement in PFS in the olaparib-treated group relative to the placebo group (hazard ratio (HR) 0.35, 95% confidence interval (CI) 0.25–0.49; *p* < 0.0001) [3]. The Phase III SOLO2 (Study of OLaparib in Ovarian cancer) trial (NCT01874353) of olaparib tablets as maintenance monotherapy in patients with platinum-sensitive HSROC and a *BRCA1/2* mutation (*BRCA*m) revealed significantly improved PFS in the olaparib-treated group compared to the placebo group (HR 0.30, 95% CI 0.22–0.41; *p* < 0.0001) [4]. 

In Korea, olaparib is approved for the maintenance treatment of patients with platinum-sensitive HSROC with *BRCA*m. The Korean Food and Drug Administration (KFDA) has permitted the use of olaparib in a maintenance setting based on the results of Study 19 since August 2015, and olaparib has been widely used clinically since January 2016—the date from which reimbursement for this treatment was initiated by the National Health Insurance Service of Korea. 

Safety data from previous trials indicate that olaparib monotherapy is generally well tolerated by recurrent ovarian cancer patients [7,8]. This suggests that the long-term use of olaparib as maintenance monotherapy is a feasible option in recurrent ovarian cancer patients. However, the real-world effectiveness of olaparib could be different from its effectiveness in clinical trials because the patient population in the real world is an unselected, general clinical practice population that includes patients with less favorable prognostic factors than patients selected for clinical trials. Here, we present our real-world experience with olaparib in patients with HSROC with *BRCA*m treated at four major tertiary institutions in Korea. 

## 2. Materials and Methods

### 2.1. Patients and Study Design

This was a retrospectively reviewed observational study of Korean women with HSROC with *BRCA*m who were treated with olaparib maintenance therapy at four institutions in Korea (Samsung Medical Center, Yonsei University Severence Hospital, National Cancer Center, and Seoul National University Hospital) between January 2016 and December 2018. In general, eligibility criteria were designed to recruit a patient population similar to the one enrolled in Study 19. All patients older than 18 years with platinum-sensitive histologically confirmed HSROC with *BRCA*m were eligible if they had received at least two previous lines of platinum-based chemotherapy, were in objective response (either complete response or partial response) to their most recent regimen, and had platinum-sensitive disease following their penultimate line of chemotherapy. All patients were monitored until December 2018 for survival and discontinuation of olaparib. This study was approved by the institutional review boards of the participating centers in accordance with the Declaration of Helsinki and the International Conference on Harmonization Good Clinical Practice guidelines (IRB No., Samsung Medical Center 2018-11-156-001; Yonsei University Severence Hospital 4-2019-0440; National Cancer Center NCC2019-0002; Seoul National University Hospital H-1812-022-991).

### 2.2. Treatment and Study Assessments

Olaparib maintenance monotherapy (400 mg bid, capsule formulation) was administered orally and continued until disease progression if toxicities were manageable. Modification of olaparib dosage was performed at the clinicians’ discretion. Patients were advised to visit every month for prescriptions, symptom check-up, and laboratory tests (complete blood count with white blood cell differential counts, liver function test, renal function test, and CA-125) and every 3 months for tumor assessment (imaging studies, mostly computed tomography scan) until objective disease progression or intolerable toxicities. Safety and tolerability were assessed throughout the follow-up period by reviewing medical records of adverse events (AEs; graded using CTCAE v5.0), physical examination results, vital signs, and laboratory findings. Endpoints of this study were safety, tolerability, duration of, and reasons for discontinuing therapy and objective response, PFS, and overall survival (OS). PFS was defined as the interval between the day olaparib was started and the first radiologically documented progression of disease or last follow-up. OS was defined as the interval between the day olaparib started and death from any cause or last follow-up. Best overall response to olaparib maintenance was defined as the best response recorded from the start of treatment until disease progression/recurrence in patients with measurable or evaluable disease. Only patients with measurable disease and a PR at entry could achieve a best overall response of complete remission (CR) or partial remission (PR). Confirmation of response (CR or PR) was performed at the next scheduled response evaluation criteria in solid tumors (RECIST) assessment and ≥ 4 weeks following the date that the criteria for response were first met [9].

### 2.3. Statistical Analysis

Summary statistics were used to describe patient characteristics. Median (range) or mean (standard deviation) were used for continuous variables. Categorical variables are presented as frequency (percentage). The Kaplan–Meier method was used to estimate PFS and OS. All statistical analyses were accomplished using IBM SPSS ver. 21.0 (IBM Corp., Armonk, NY, USA).

## 3. Results

Between January 2016 and December 2018, 100 HSROC patients with *BRCA1/2* mutation were treated with olaparib maintenance therapy at four institutions (44 patients from Samsung Medical Center, 25 from Yonsei University Severence Hospital, 19 from the National Cancer Center, and 12 from Seoul National University Hospital). Data cutoff for final analysis was December 2018; the median duration of follow-up was 10.2 months (range, 1.0–35.7 months). 

### 3.1. Patient Characteristics

Baseline characteristics are summarized in Table 1. Ninety-four patients had germline *BRCA* mutations; 69 patients (69.0%) had *BRCA1* mutations and 24 patients (24.0%) had *BRCA2* mutations. Somatic *BRCA* mutations were present in six patients (6.0%): three patients had *BRCA1* mutations, one had *BRCA2* mutations, and two had *BRCA1/2* mutations. Altogether, 71 patients (71%) had *BRCA1* mutations, 23 patients (23%) had *BRCA2* mutations, and three patients (3%) had *BRCA1/2* mutations. International Federation of Gynecology and Obstetrics (FIGO) stage IIIC at initial diagnosis was most common (47.0%) among the patients. Most patients had undergone two chemotherapy regimens before olaparib (*n* = 63, 63.0%), while some patients had undergone three regimens (*n* = 26, 26.0%). Four patients had undergone five or more previous regimens. Platinum-free interval was > 12 months in 68 patients (68.0%) and 6–12 months in 32 patients (32.0%) (median 14.6 months (6.0~86.3 months)). There were 16 patients (16.0%) with previous bevacizumab exposure. In terms of objective response to most recent chemotherapy, complete response was shown in 46 patients (46.0%) and partial response in 53 patients (53.0%).

### 3.2. Treatment Exposure

At the median follow-up duration of 10.2 months (range, 1.0–35.7 months), 62 patients were still being treated with olaparib. Percentages for the number of olaparib users per total patients for different time periods are shown in Table 2. Sixty-two percent of patients (53/85) were using olaparib after 6 months, 53.5% (23/43) after 12 months, and 50.0% (4/8) after 24 months. Olaparib was discontinued due to disease progression in 34 patients (34.0%) and unacceptable toxicity in four (4.0%) (grade 3 neutropenia and associated complications in one patient, grade 4 anemia in one, grade 2 nausea/vomiting in one, and grade 3 soft tissue infection in one patient). Characteristics of the four long-term responders on olaparib maintenance are shown in Appendix A. There were no significant differences in clinicopathological characteristics between long-term and non-long-term responders at this point.

### 3.3. Efficacy

In terms of best overall response to olaparib maintenance in 53 patients with measurable or evaluable disease (partial remission, PR) from most recent chemotherapy, complete remission (CR) was shown in 12 patients (22.6%), partial remission (PR) in four (7.5%), and stable disease in 33 (62.3%) (Table 2). Among the total of 100 patients, 37 recurrences (37.0%) and five deaths (5.0%) were observed during the follow-up period. Despite the short follow-up period, we derived Kaplan–Meier estimates of progression-free survival and overall survival from the start of olaparib maintenance therapy (Figure 1). Median PFS from the start of olaparib maintenance therapy was 14.6 months (95% CI 9.65~19.61), with a 24 month PFS of 42.4% (Figure 1A). Median OS was not reached, and 24 month OS was 82.1% from Kaplan–Meier estimates (Figure 1B). Additionally, characteristics of 37 patients with recurrence after olaparib maintenance therapy are shown in Appendix A.

### 3.4. Safety

AEs are shown in Table 3. A total of 64 AEs (64.0%) comprising 37 hematological (37.0%) and 27 non-hematological (27.0%) adverse events were observed. The most common hematological AE was anemia (24.0%), while nausea/vomiting (13.0%) was the most common non-hematological AE. Twenty-five episodes of grade 3 or 4 AEs were documented: 14 cases (14.0%) of anemia, seven cases of neutropenia (7.0%), two cases of thrombocytopenia (2.0%), one case of oral mucositis (1.0%), and one case of soft tissue infection (1.0%). Only two cases with grade 3 or more non-hematologic AEs were observed. Dose reduction (DR) was performed in 41 cases (41.0%), and the drug was discontinued in eight (6.0%) cases. Continuation without dose modification was documented in 15 cases (15%). There were no cases with drug interruption. Median time to onset of the first AE was 1.9 months (range 0.1–8.8 months). 

Time to first AE (event rate per patient-years) is shown in Figure 2. Patients mostly experienced AEs for the first time within a few months of the start of olaparib treatment, and it was rare for AEs to initially develop after 6 months on olaparib (Figure 2A). Anemia (Figure 2B) and nausea/vomiting (Figure 2C) showed a similar pattern to that of total AEs. Prevalence plots also showed early onset of AEs after olaparib treatment initiation (Figure 2D,E,F). The numbers of AEs with time to first event are shown in Appendix A. Forty-eight of 64 AEs (75.0%) occurred within the first 3 months from the start of olaparib treatment (Appendix A A). AEs by grade with time to first event are shown in Appendix A B. Hematologic and non-hematologic AEs had a similar pattern, with mostly early onset within 3 months of treatment initiation (70.3% and 81.5%, respectively) (Appendix A).

Information regarding 41 cases with dose reduction (DR) due to AEs is provided in Table 4. In 20 cases (48.8%), 25% DR was performed, while in the remaining 21 cases (51.2%), 50% DR was performed. In eight cases (19.5%), the dose was normalized after the AEs resolved, while dose reduction was maintained in 28 cases (68.3%). In five cases (12.2%), DR due to AEs led to discontinuation of olaparib. Median DR duration was 6.5 months (range 1.0–32.4 months).

## 4. Discussion

Our real-world experience of the safety and effectiveness of olaparib maintenance treatment in HSROC patients with *BRCA1/2* mutation is consistent with that reported in previous studies [3,4,10] with tolerable toxicity profiles. Our results can be compared to those of Study 19, a phase II trial of olaparib capsules as maintenance monotherapy in patients with platinum-sensitive HSROC, as we used the same indications and regimen (capsule 400 mg bid). The use of olaparib in Korea was permitted based on the results of Study 19, and it is mandatory for physicians in Korea to follow the recommendations and guidelines of the Korea Health Insurance Review and Assessment Service (KHIRA). 

There were some differences between olaparib-treated patients with *BRCA*m (*n* = 74) in Study 19 and patients in our study [10]. Median patient age was greater in Study 19 (57.5 years vs. 54.0 years) and we only included patients with ovarian cancer, while Study 19 included patients with fallopian tube and primary peritoneal cancer (12%). Furthermore, our study included dominantly Asian patients in contrast to Study 19. *BRCA* mutation status was similar between Study 19 (*BRCA1* 65%; *BRCA2* 35%; *BRCA1/2* 2%) and our study (*BRCA1* 72%; *BRCA2* 25%; *BRCA1/2* 3%). In Study 19, 18 of 136 patients with *BRCA*m (13.2%) had definite somatic *BRCA*m [10], while we had only six patients (6.0%) with somatic *BRCA* mutation. Platinum-free interval and objective response to the most recent chemotherapy were similar in the two studies [3,10]. Despite an insufficient follow-up period in our study, median PFS was similar (14.6 months compared to 11.2 months in patients with *BRCA*m in Study 19). There were also differences in observed numbers of AEs between our study and Study 19. Among patients with *BRCA*m in Study 19, 97% had AEs. The most common non-hematologic AE was nausea (73%), while the most common hematologic AE was anemia (26%). In our study, 64 total episodes (64.0%) of AEs were observed, with 13 cases (13.0%) of nausea/vomiting and 24 cases (24.0%) of anemia. The percentage of anemia was similar in the two studies (26% vs. 24%), but grade ≥ 3 anemia was observed in only 5% of patients in Study 19 compared to 14% of our patients. 

*BRCA* status, the number of previous chemotherapy regimens, and platinum-free intervals in our study were similar to those reported by the SOLO2 study [4]. Median age was also similar, but our age range was wider (56 years (range 51–63) vs. 53 years (29–79)). In SOLO2, the 24 month PFS was 43%, while ours was 42.4%. Higher numbers of AEs were reported in the SOLO2 study than in our study. In SOLO2, ninety-eight percent of total patients had AEs with nausea (76%), most common for non-hematologic AE, and anemia (43%) for hematologic AE. Anemia and nausea were the two most common AEs in our study as well, but fewer patients in our study than in the SOLO2 study experienced these AEs. Differences between our findings and those of Study 19 and SOLO2 might be due to differences in tablet and capsule compositions. In Study 19, which administered olaparib in capsule form, 68.4% of patients experienced nausea and 16.9% experienced anemia [3] compared to 76% with nausea and 43% with anemia in the SOLO2 study that administered olaparib in tablet form. Currently, there are no established guidelines for transitioning patients from capsules to tablets for olaparib [11]. However, there should be further studies of capsules and tablets to determine their relative efficacy and safety.

Differences in AEs between our study and Study 19 might have been caused by different treatment patterns in a real-world setting and biases caused by the retrospective nature of the study. Randomized controlled trials (RCTs) should adhere strictly to the study protocol, but in the real world, treatment can vary depending on the physician [12]. In Study 19, treatment was interrupted for any grade 3 or 4 AE that was considered related to treatment; if the toxicity resolved or decreased in severity to grade 1, treatment was restarted with a reduction in dose to 200 or 100 mg twice daily. In our study, physicians usually reduced the dose without interruption, with 25% DR in 49% of patients with AEs rather than an immediate 50% DR. In Study 19, dose interruption occurred in 36% of patients overall, including 42% of patients in the olaparib-treated group. In our data, the dose was reduced in 36% patients, with no drug interruptions. Eight patients had dose re-escalation after dose reduction compared to 28 patients who maintained their reduced dose. There is the possibility that maintenance on a reduced dose may account for the lower rate of AEs in our study. Additionally, in a real-world setting, it is difficult for physicians to record all symptoms unless informed by the patient, which could also have contributed to our lower rate of AEs. Our study is based on retrospectively collected data, and there is the possibility of bias caused by incomplete data collection.

Real-world data are important as they involve circumstances and conditions outside the scope of highly controlled clinical trial settings [13]. RCTs focus on the efficacy of a drug, whereas real-world data can provide information about feasibility, effectiveness, safety, epidemiology, and/or the costs of treatment related to the drug [14]. Although data from RCTs have the highest reliability, follow-up observations from completed RCT research are important as they reflect actual clinical aspects, and RCT and real-world data can complement one another.

To the best of our knowledge, this study is the first to describe a real-world experience with olaparib maintenance in HSROC patients with *BRCA*m. Inclusion criteria, treatment, and follow-up protocols were relatively consistent among institutions and physicians because medical practices in Korea are performed under the guidelines and recommendations of the KHIRA. Olaparib has to be prescribed every month based on laboratory tests by physician appointment, and image evaluations (mostly CT scans) are required every 3 months.

There are several potential limitations to the current study. First, it was based on retrospectively collected data. Therefore, our study may have had biases, especially due to incomplete data collection, and the lack of specific protocols and safety profiles in a real-world setting may have interfered with the accurate interpretation of data. The use of multi-institutional study data could also be considered a limitation of our study because, in a retrospective setting, multiple physicians from different institutions may not have used identical management protocols for each clinical situation. Also, the patient number was not large enough for more specific subgroup analysis, and observation time was not long enough to determine survival outcomes.

## 5. Conclusions

In summary, results from this multi-institutional, retrospective, observational study indicate that olaparib maintenance treatment is both effective and safe for HSROC patients with *BRCA1/2* mutation in a real-world clinical practice, as has been demonstrated in clinical trials. However, a longer follow-up period is needed for survival analysis.

## Figures and Tables

**Figure 1 jcm-08-01920-f001:**
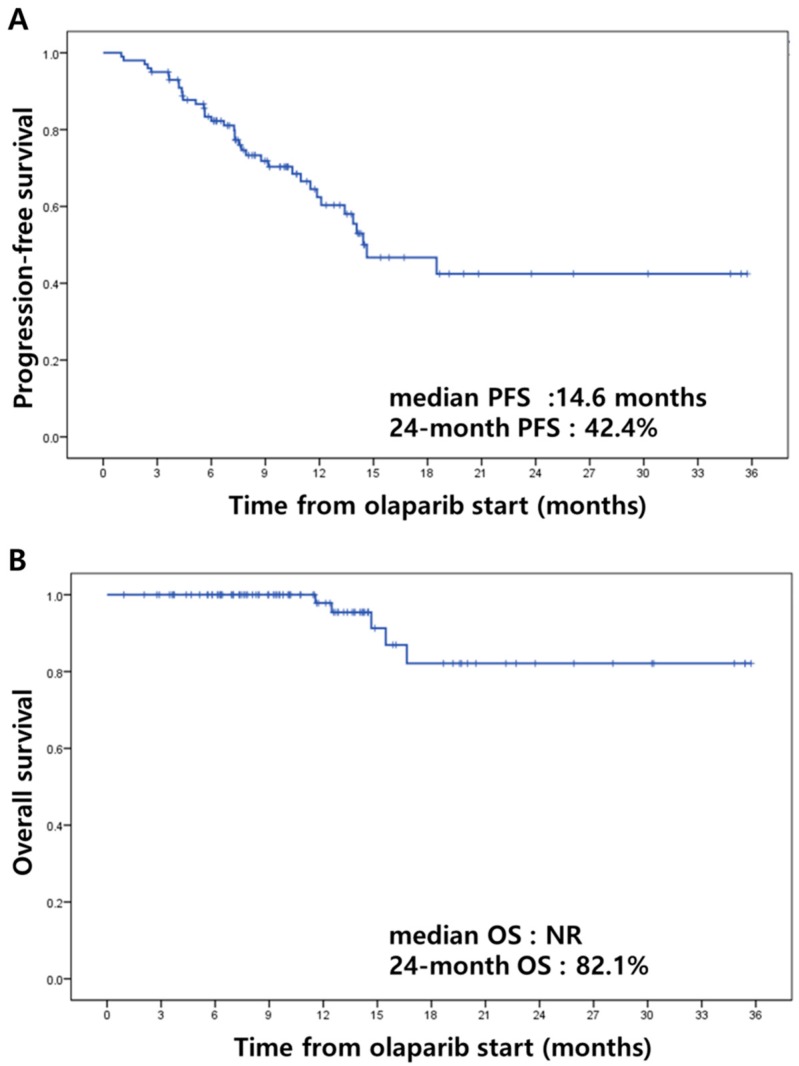
Kaplan–Meier estimates of progression-free survival (**A**) and overall survival (**B**). (NR: not reached).

**Figure 2 jcm-08-01920-f002:**
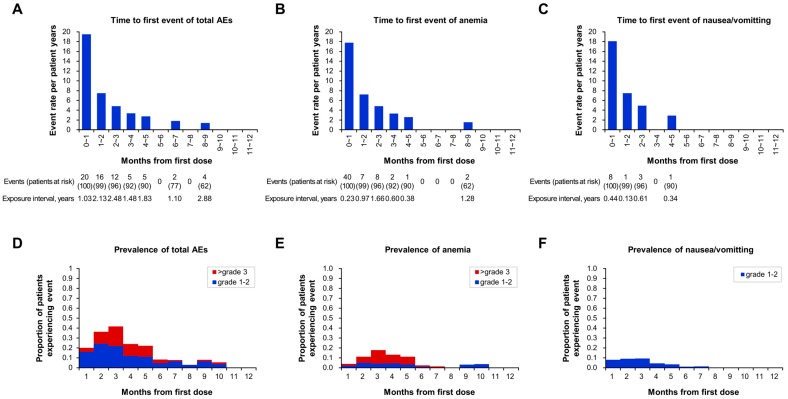
Characteristics of adverse events (AEs): time to first event (event rate = number of first events/exposure during time interval), (**A**) total AEs, (**B**) anemia, and (**C**) nausea/vomiting and prevalence by month and grade of (**D**) total AE, (**E**) anemia, and (**F**) nausea/vomiting in olaparib-treated patients.

**Table 1 jcm-08-01920-t001:** Basal characteristics of patients (*N* = 100).

Characteristics	*N* = 100
Age, years	
Median (Range)	54 (29~79)
Initial FIGO stage, *n* (%)	
IA	1 (1.0)
IB	1 (1.0)
IC	6 (6.0)
IIA	1 (1.0)
IIB	3 (3.0)
IIIA	2 (2.0)
IIIB	6 (6.0)
IIIC	47 (47.0)
IV	33 (33.0)
Initial residual status, *n* (%)	
No residual	41 (41.0)
0.1~1 cm	45 (45.0)
>1 cm	13 (13.0)
Initial CA-125 level	720.5 (14.2~11,552.2)
Concurrent breast cancer, *n* (%)	18 (18.0)
Family history of breast and ovarian cancer, *n* (%)	23 (23.0)
Platinum-free interval Duration, *n* (%)	
6–12 months	32 (32.0)
>12 months	68 (68.0)
Median (range), months	14.6 (6.0~86.3)
Objective response to most recent chemotherapy, *n* (%)	
Complete	46 (46.0)
Partial	53 (53.0)
Unknown	1 (1.0)
Number of previous chemotherapy regimen, *n* (%)	
2	63 (63.0)
3	26 (26.0)
4	7 (7.0)
≥5	4 (4.0)
Median (range)	2 (2~13)
Previous bevacizumab exposure, *n* (%)	16 (16.0)
*BRCA* mutation status, *n* (%)	
Germline	
*BRCA1*	69 (69.0)
*BRCA2*	24 (2.0)
*BRCA1/2*	1 (1.0)
Somatic	
*BRCA1*	3 (3.0)
*BRCA2*	1 (1.0)
*BRCA1/2*	2 (2.0)

FIGO: International Federation of Gynecology and Obstetrics.

**Table 2 jcm-08-01920-t002:** Outcomes of olaparib maintenance in high-grade serous recurrent ovarian cancer patients with *BRCA1/2* mutation.

Outcomes	
Observation period after olaparib	
Median (range), months	10.2 (1.0~35.7)
Olaparib maintenance period	
Olaparib users/total patients at time period (%)	
≥6 months	53/85 (62.4)
≥12 months	23/43 (53.5)
≥24 months	4/8 (50.0)
Best overall response with olaparib in patients with previous PR from most recent chemotherapy, *n* (%)	*N* = 53
CR	12 (22.6)
PR	4 (7.5)
SD	33 (62.3)
PD	2 (3.8)
NE	2 (3.8)
Type of dose modification due to AE, number of patients (%)	*N* = 100
Dose reduction	36 (36.0)
Drug discontinuation	4 (4.0)
Treatment applied other than dose reduction	11 (11.0)
Recurrence event, *n* (%)	37 (37.0)
Death event, *n* (%)	5 (5.0)

CR: complete remission, PR: partial remission, SD: stable disease, PD: progression of disease, NE: not evaluable, and AE: adverse event.

**Table 3 jcm-08-01920-t003:** Adverse events (AEs).

	Total episodes, *n* (%)	Grade			Characterization of AEs	Management		
		Grade 1–2, *n* (%)	Grade 3, *n* (%)	Grade 4, *n* (%)	Median time to onset of first event, months, median (range)	Dose reduction, *n* (%)	Drug discontinuation, *n* (%)	Treatment applied with continuation (without dose reduction), *n* (%)
**Any adverse event**	64 (64.0)	39 (39.0)	19 (19.0)	6 (6.0)	1.9 (0.1~8.8)	41 (41.0)	8 (6.0)	15 (15.0)
**Hematological**	37 (37.0)	14 (14.0)	17 (17.0)	6 (6.0)	2.1 (0.1~8.8)	22 (22.0)	6 (6.0)	9 (9.0)
Anemia	24 (24.0)	10 (10.0)	8 (8.0)	6 (6.0)	2.1 (0.7~2.8)	15 (14.0)	3 (3.0)	6 (6.0)
Neutropenia	7 (7.0)		7 (7.0)		1.8 (0.1~8.8)	4 (4.0)	2 (2.0)	1 (1.0)
Thrombocytopenia	4 (4.0)	2 (2.0)	2 (2.0)		2.5 (0.1~3.5)	3 (3.0)	1 (1.0)	
Serum ALT/AST elevated	2 (2.0)	2 (2.0)			4.6			2 (2.0)
**Non hematological**	27 (27.0)	25 (25.0)	2 (2.0)		1.6 (0.4~6.7)	19 (19.0)	2 (2.0)	6 (6.0)
Nausea/vomiting	13 (13.0)	13 (13.0)			2.5 (0.4~4.1)	9 (9.0)	1 (1.0)	3 (3.0)
Fatigue	6 (6.0)	6 (6.0)			1.9 (0.5~6.7)	5 (5.0)		1 (1.0)
Oral mucositis	2 (2.0)	1 (1.0)	1(1.0)		4.2	2 (2.0)		
Peripheral neuropathy	2 (2.0)	2 (2.0)			1.4	1 (1.0)		1 (1.0)
Urticaria	1 (1.0)	1 (1.0)			0.8			1 (1.0)
Dizziness	1 (1.0)	1 (1.0)			4.7	1 (1.0)		
Headache	1 (1.0)	1 (1.0)			2.5	1 (1.0)		
Soft tissue infection	1 (1.0)		1(1.0)		1.7		1 (1.0)	

AE: adverse event, AST: aspartate aminotransferase, and ALT: alanine aminotransferase. Patients may have experienced more than one episode of adverse effect.

**Table 4 jcm-08-01920-t004:** Dose reduction (DR) pattern.

	Number of Episodes, *n* (%)	Reduction Dose, *n* (%)		After Dose Reduction Initiation, *n* (%)			Median Time to Onset of First Event, Months, Median (Range)	DR Duration, Months, Median (Range)
		25% DR	50% DR	Dose normalized after AE resolved	Maintain DR	Discontinuation after DR		
**Any adverse event**	41 (100.0)	20 (48.8)	21 (51.2)	8 (19.5)	28 (68.3)	5 (12.2)	1.9 (0.1~6.7)	6.5 (1.0~32.4)
Grade 1–2	24 (100.0)	15 (62.5)	9 (37.5)	3 (12.5)	19 (79.2)	2 (8.3)	1.9 (0.1~6.7)	5.6 (1.0~32.4)
Grade 3	12 (100.0)	3 (25.0)	9 (75.0)	4 (33.3)	5 (41.7)	3 (25.0)	1.6 (0.1~4.6)	5.1 (1.0~30.8)
Grade 4	5 (100.0)	2 (40.0)	3 (60.0)	1 (20.0)	4 (80.0)	0	2.1 (1.7~2.8)	8.1 (1.2~11.1)
**Hematological**	22 (100.0)	8 (36.4)	14 (63.6)	6 (27.3)	12 (54.5)	4 (18.2)	1.8 (0.1~4.6)	5.4 (1.0~30.8)
Grade 1–2	5 (100.0)	3 (60.0)	2 (40.0)	1 (20.0)	3 (60.0)	1 (20.0)	1.8 (0.1~3.7)	4.0 (1.0~8.3)
Grade 3	12 (100.0)	3 (25.0)	9 (75.0)	4 (33.3)	5 (41.7)	3 (25.0)	1.6 (0.1~4.6)	5.1 (1.0~30.8)
Grade 4	5 (100.0)	2 (40.0)	3 (60.0)	1 (20.0)	4 (80.0)	0	2.1 (1.7~2.8)	8.1 (1.2~11.1)
**Non hematological**	19 (100.0)	12 (63.2)	7 (36.8)	2 (10.5)	16 (84.2)	1 (5.3)	1.9 (0.4~6.7)	7.2 (1.0~32.4)
Grade 1–2	19 (100.0)	12 (63.2)	7 (36.8)	2 (10.5)	16 (84.2)	1 (5.3)	1.9 (0.4~6.7)	7.2 (1.0~32.4)


DR: dose reduction and AE: adverse event. Patients may have experienced more than one episode of adverse effect.

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
