# Peer review of "Real-World Experience of Olaparib Maintenance in High-Grade Serous Recurrent Ovarian Cancer Patients with BRCA1/2 Mutation: A Korean Multicenter Study"

_jcm, 2019, doi:10.3390/jcm8111920_

Round 1
Reviewer 1 Report
The work shows the safety and efficacy of olaparib maintenance treatment. The results are in line with what is reported by other case studies. It would be useful to report on the recurrence of recurrences of this patient setting and to understand if olaparib therapy can influence the choice of second-instance surgical treatment (Eur J Surg Oncol. 2019 Jun 13. Prognostic factors value of germline and somatic brca in patients undergoing surgery for recurrent ovarian cancer with liver metastases)
Author Response
Reviewer 1
The work shows the safety and efficacy of olaparib maintenance treatment. The results are in line with what is reported by other case studies. It would be useful to report on the recurrence of recurrences of this patient setting and to understand if olaparib therapy can influence the choice of second-instance surgical treatment (Eur J Surg Oncol. 2019 Jun 13. Prognostic factors value of germline and somatic brca in patients undergoing surgery for recurrent ovarian cancer with liver metastases)
-> In our study, included patients are with recurrent ovarian cancer patients with at least two previous lines of platinum based chemotherapy, and patients who used olaparib as maintenance setting. It is difficult to show recurrence pattern of patients with multiple recurrence history, and generally, ovarian cancer patients with multiple recurrence are not indicated for surgical treatment. For additional information, we added supplementary data of 37 patients with recurrence after olaparib maintenance (S2). (Page 5, line 157)

Reviewer 2 Report
This study is limited due to the relatively small number of patients and the short follow-up time. Subgroup analyses and some survival analyses are therefore not possible. These limitations reduce the impact of the report, and the authors acknowledge these limitations. However, this appears to be the first report of real-world use of maintenance olaparib in recurrent HGSOC, and is certainly an important complement to previous clinical trials such as Study 19 and SOLO2. It can serve as a good first look at maintenance olaparib therapy, with additional follow-up studies to come.
I have only minor concerns about text editing and figure formatting, which I will outline below. Please make the following edits and corrections, especially in regards to use of abbreviations and improving the appearance and readability of the figures:
There are several places in the article where an abbreviation is used without fully spelling out what the abbreviation means. For example, RECIST is used on page 3, line 104 of the Materials and Methods. Please go through the entire text and be sure to introduce the full meaning of all abbreviations before using them. While the primary audience of this article is physicians, and they are familiar with these terms, some readers will be unfamiliar and therefore these terms need to be introduced properly. The text on Figure 1 and Figure 2 is much too small. These figures should be re-formatted to make the text larger. Ideally, they should be redone in GraphPad Prism or some other program to make them more visually appealing and easy to read. If the authors only have Excel, then please reformat to improve ease of reading. The caption of Figure 2 includes an explanation of panel B-F, but not panel A. A quick rewrite is needed. Page 11, line 2: "tolaparib" should be "olaparib" Page 11, lines30-31: The sentence reads "Ninety-eight percent of total patients had nausea (76%)... while 43% had anemia (43%)." The first part doesn't make sense. It can't be both 98% and 76%. Was it 98 total patients that had nausea? Please rewrite and fix. The second part is redundant (43% is shown twice in a row). Please edit to tell the total number of patients with anemia, then show the percentage. The use of the term "RCT" on page 11, line 41. The authors have not introduced what RCT stands for. Again, please go through line by line and introduce a term before using the abbreviation.
Author Response
Reviewer 2
This study is limited due to the relatively small number of patients and the short follow-up time. Subgroup analyses and some survival analyses are therefore not possible. These limitations reduce the impact of the report, and the authors acknowledge these limitations. However, this appears to be the first report of real-world use of maintenance olaparib in recurrent HGSOC, and is certainly an important complement to previous clinical trials such as Study 19 and SOLO2. It can serve as a good first look at maintenance olaparib therapy, with additional follow-up studies to come.
I have only minor concerns about text editing and figure formatting, which I will outline below. Please make the following edits and corrections, especially in regards to use of abbreviations and improving the appearance and readability of the figures:
There are several places in the article where an abbreviation is used without fully spelling out what the abbreviation means. For example, RECIST is used on page 3, line 104 of the Materials and Methods. Please go through the entire text and be sure to introduce the full meaning of all abbreviations before using them. While the primary audience of this article is physicians, and they are familiar with these terms, some readers will be unfamiliar and therefore these terms need to be introduced properly.
->Thank you for good point. We added full terminology to manuscript. (page 3, line 104, 123)
The text on Figure 1 and Figure 2 is much too small. These figures should be re-formatted to make the text larger. Ideally, they should be redone in GraphPad Prism or some other program to make them more visually appealing and easy to read. If the authors only have Excel, then please reformat to improve ease of reading.
->We changed figure 1 and 2 for better visualization.
The caption of Figure 2 includes an explanation of panel B-F, but not panel A.
->We added caption for Figure 2-A
A quick rewrite is needed. Page 11, line 2: "tolaparib" should be "olaparib" Page 11, lines30-31:
->We changed to olaparib.
The sentence reads "Ninety-eight percent of total patients had nausea (76%)... while 43% had anemia (43%)." The first part doesn't make sense. It can't be both 98% and 76%. Was it 98 total patients that had nausea? Please rewrite and fix. The second part is redundant (43% is shown twice in a row). Please edit to tell the total number of patients with anemia, then show the percentage.
->We are sorry to make confusion. Mentioned part was corrected. : In SOLO2, ninety-eight percent of total patients were shown with AEs with nausea (76%) for most common for non-hematologic AE and anemia (43%) for hematologic AE.(page 11,line 31)
The use of the term "RCT" on page 11, line 41. The authors have not introduced what RCT stands for. Again, please go through line by line and introduce a term before using the abbreviation.
-> We added full terminology to manuscript. (page 11 , line 43)

Round 2
Reviewer 1 Report
good work